# TempODEGraphNet: predicting user churn using dynamic social graphs and neural ODEs

**Minseop Lee¤, Jiyoung Woo** [ID]¤*

Dept. ICT convergence, University of Soonchunhyang, Asan, South Korea

¤ 22-15, Soonchunhyang-ro, Sinchang-myeon, Asan-si, Chungcheongnam-do, Republic of Korea
* jywoo@sch.ac.kr

**Data availability statement:** The dataset can be accessed at https://doi.org/10.6084/m9.figshare.28233857.v1

## Abstract

Research on user churn prediction has been conducted across various domains for a long time. Among these, the gaming domain is characterized by its potential for diverse types of interactions between users. Due to this characteristic, many studies on churn prediction have considered the relationships between users and have primarily applied social network analysis. Recently, the use of Graph Neural Networks (GNNs) has been actively applied. However, existing studies utilizing GNNs have limitations as they use static graphs that do not effectively capture the dynamic nature of interactions that change over time. This study addresses these limitations by proposing a dynamic graph model for predicting user churn in games based on user interactions. Data are sourced from 10,000 users of 'Blade & Soul' by NCSOFT. The proposed model effectively captures changes in user behavior over time and predicts user churn with a focus on interactions among users. Experimental results reveal that the proposed model achieves a higher F1 score compared with conventional algorithms and static graph models. Dynamic graphs more accurately reflect changes in user behavior compared with static graphs, particularly in domains with active interactions such as massively multiplayer online role-playing games. This work highlights the significance of user churn prediction in the gaming industry and demonstrates the effectiveness of the predictive models that use dynamic graphs.

## Introduction

In the companies that acquire economic profit in exchange for providing services to consumers, the number of users strongly affects the revenue. Consequently, considerable efforts are made toward attracting new users or retaining existing users. As research has revealed that retaining existing users is more economically beneficial than attracting new users, methods for the early detection of user churn have been extensively investigated [1].

**Funding:** Our work was financially supported by Soonchunhyang University and the Regional Innovation Strategy (RIS) through the National Research Foundation of Korea (NRF) by the Ministry of Education (MOE) under Grant 2021RIS-004. The funders had no role in study design, data collection and analysis, decision to publish, or preparation of the manuscript.

**Competing interests:** No authors have competing interest.

Early studies on user churn prediction experienced challenges owing to limited data collection and analysis techniques. Predictive models primarily relied on simple rule-based approaches because of small and restrictive datasets, e.g., "if certain metrics decrease below a certain level, there is a risk of churn." These approaches had limitations in understanding and predicting the complex patterns of user churn. Despite these limitations, these early studies provided the foundation for future research by emphasizing the importance of understanding user behavior and identifying at-risk users for churn [2,3].

The widespread adoption of machine learning techniques in businesses has led to significant advancements in the field of user churn prediction. Machine learning algorithms, such as decision trees, logistic regression, naive Bayes, and support vector machines, have been used for churn prediction. These algorithms can handle larger and more complex datasets and detect subtle patterns within data to enhance prediction accuracy. However, the performance of these models strongly relies on feature engineering and data quality. A data-centric approach has been introduced based on machine learning models to reduce reliance on manual rule-based definitions. This has enabled businesses to effectively respond to user behavior and accurately predict churn [4,5].

There has been a shift in the paradigm of user churn prediction because of the increasing use of data science. Companies are focusing on collecting and utilizing a wide range of user data to enhance the accuracy of churn prediction models. This involves a comprehensive examination of user characteristics including factors such as purchase history, activity patterns, geographic information, gender, and age. This has led to a data-rich environment that reveals hidden correlations and causal relationships among user attributes, thereby enabling models to make accurate predictions [6,7]. The utilization of deep learning and neural network technologies has led to major changes in the field of user churn prediction. In particular, neural network models, such as recurrent neural networks and long short-term memory (LSTM) networks, have garnered attention because of their ability to comprehend complex user behavior patterns [8]. These technological advancements have opened new avenues for churn prediction models. Neural networks can model intricate relationships within sequential data. They can be used to detect and predict considerably more complex and dynamic user churn patterns compared with previous methods [9,10].

In the gaming domain, the massively multiplayer online role-playing game (MMORPG) genre allows for various activities that are similar to real life and strong interactions among users. Numerous studies on churn prediction have used features generated from user interactions such as party records, transaction records, friend records, and chat logs [11,12]. In particular, Tao et al. utilized graph convolution Network (GCN) models, where a graph was formed using friend records, and trained a model using this graph [13,14].

The key difference between using graph neural networks and conventional social network analysis centrality metrics as features is that the former can incorporate structural information such as user interactions and in-game positions along with individual user features. Centrality metrics provide only disconnected information, whereas graph neural networks provide a more comprehensive understanding by considering interactions among users and roles within a community.

In this study, we propose a methodology that utilizes dynamic graphs, where the edge information between nodes changes over time, instead of observing various networks, such as user-party and user-transaction networks, at a single time point. This approach accurately reflects changes in user behavior patterns and interactions over time and captures the evolving dynamics of user interactions. This study contributes to the research on user churn prediction in online games in the following two main aspects:

1. We introduce a model where dynamic graphs generated at multiple timestamps are integrated through bidirectional LSTM (Bi-LSTM) [15]. This model utilizes embeddings generated by GCN layers from each timestamp's graph as inputs to Bi-LSTM, thus enabling the sensitive capture of changes in user behavior patterns over time.

2. We apply a neural ordinary differential equation (ODE), which provides approximations that are closer to primitive functions compared with neural network training methods, to user churn prediction in online games for the first time to enhance the consistency of model performance [16].

## Related works

### Paper review on game user churn prediction

User churn prediction models have improved with the advancement of artificial intelligence and computing systems. We conducted a review to understand the techniques or models used in various studies, key features, and utilization of social interaction data for churn prediction as shown in Table 1.

Kawale et al. [17] applied social network analysis to networks created from user party records in the gaming domain. They generated centrality features from these networks and used an AdaBoost model to classify churn and retention. The model was trained using the centrality features derived from party networks and conventional personal features such as the playtime, login frequency, and payment records. Kim et al. [26] applied social network analysis to perform community detection and calculate centrality. They classified churn and retention using a model with custom-defined learning methods and structures. The model

**Table 1. Summary of related works on game user churn prediction.**

| Year | Author | Algorithm | Data |
|------|--------|-----------|------|
| 2009 | Kawale et al. [17] | Apply social network analysis, perform classification using AdaBoost | Party, playtime, log-in frequency, payment, etc. |
| 2016 | Tamassia et al. [18] | Apply social network analysis, perform classification using hidden Markov model | Party, playtime, log-in frequency, payment, etc. |
| 2017 | Kim et al. [19] | Use LSTM to efficiently respond to over time series dataset | Playtime, log-in frequency, payment, etc. |
| 2017 | Bertens et al. [20] | Conduct survival analysis centered on residual users through conditional inference survival ensemble | Playtime, log-in frequency, payment, etc. |
| 2018 | Moon et al. [21] | Developed an efficient churn prediction model using MMORPG social activity data | Quest, guild, trade, etc. |
| 2018 | Seo et al. [22] | Categorize MMORPG users by social activity tendencies within guilds | Guild, quest etc. |
| 2020 | Lee et al. [23] | Apply social network analysis, perform classification using XGBoost, gradient boosting machine, random forest, etc. | Party, trade, playtime, log-in frequency, payment, etc. |
| 2022 | Oskarsdottir et al. [24] | Apply social network analysis, perform classification using random forest, etc. | Friend, playtime, log-in frequency, payment, in-game trade history |
| 2023 | Tao et al. [25] | Perform classification using static GCN | Friend, chat, playtime, log-in frequency, etc. |

was trained using network centrality metrics, community metrics, and conventional personal features. Tamassia et al. [18] applied social network analysis to networks generated from user party records to create and use centrality features. They performed time-series classification using a hidden markov model. The model was trained using centrality features generated from the party network and conventional personal features. Kim et al. [19] performed churn and retention classification using an LSTM model, which was effective for modeling complex relationships in time-series datasets. The model was trained using conventional personal features. Lee et al. [23] and Oskarsdottir et al. [24] applied social network analysis to networks generated from party, trade, and friend interactions to create centrality features. They utilized models such as random forests to classify churn and retention. The models were trained using centrality features derived from social network analysis and conventional personal features. Moon et al. [21] investigated the impact of social activities in MMORPGs on user churn using actual game data and features extracted from prechurn activity logs to develop an efficient churn prediction model with reduced feature dimensions. Seo et al. [22] classified MMORPG users based on their social activity tendencies within guilds, analyzed churn rates and reasons for churn among categorized groups, and proposed a framework to predict churn by measuring gameplay engagement trends among these groups. Their framework used data from Aion, which is a flagship MMORPG by NCSOFT, and achieved an average precision of approximately 75%. Tao et al. [25] classified churn and retention by applying a static GCN to networks generated from friend and chat records. The feature matrix consisted of conventional personal features.

In summary, studies on user churn prediction have focused on identifying patterns of changes in user behavior over time. Across various domains, churn has been predicted on the basis of social activities by utilizing the principle of homophily, where contact among similar individuals occurs at a higher rate than that among dissimilar individuals [27]. In the gaming domain, studies have actively incorporated social activities such as friend interactions, trading, and chatting within games. This is mainly achieved by applying social network analysis to networks formed based on specific activities and deriving centrality features. Recent works have used GCNs, where embeddings are generated based on relationships between users, thus moving beyond simply indicating relative positions within a graph.

However, static graph models aggregate user interactions over a fixed period, resulting in a loss of temporal information. Static GCNs have inherent limitations in dynamic environments, as node states and edge connections remain fixed throughout the observation period, disregarding the temporal dynamics of user behavior. For instance, interactions between users (edges) that exist at one timestamp may disappear at the next, reflecting changes in social engagement. By collapsing these interactions into a single graph, static models overlook such fluctuations, leading to incomplete representations of user relationships. This static representation fails to capture the evolving nature of user interactions, which is crucial for predicting churn in dynamic environments such as MMORPGs.

## Backgrounds

In the early stages of deep learning, Deep neural networks(DNNs) were developed to handle regular types of data effectively. Later, Recurrent neural networks(RNNs)-based models emerged, specializing in sequential data with inherent order. RNNs shows significant potential in various machine learning tasks for time-series data. The RNN processes the data iteratively, maintaining a state that evolves by transforming prior sequences. This state acts as a memory, capturing how earlier values influence the prediction system. The input to the RNN is represented as a 2D tensor with dimensions corresponding to timesteps and input features.

At each timestep, the current input is fed into the RNN, which updates its state by combining the current input with the previously retained state. This updated state is then used to process the subsequent input, enabling the network to dynamically learn from the sequence over time. A simple RNN model combines the current input and the previous hidden state directly, so in the long-time series data, former input data is forgotten. To overcome this drawback, the Long Short-Term Memory (LSTM) [32] introduces a memory cell that maintains earlier inputs in their original form and supplies them to the RNN node when necessary. Additionally, the input is selectively passed to the hidden state, which connects to the input in the next iteration. This selective memory and state mechanism allows LSTMs to effectively capture long-term dependencies in sequential data. Bi-LSTM (Bidirectional Long Short-Term Memory) processes sequential input data in the backward direction as well as the forward direction. It enables the model to capture context from both the past and the future within the input sequence. DNNs and RNNs are not suitable for graph data that involves relational structures. This is because graph data represents relationships between nodes, a characteristic that DNNs cannot effectively capture. Moreover, graph data lacks a meaningful order among its data, making RNNs unsuitable since they process data sequentially. Graph neural networks (GNN) are specialized neural networks that are designed for graph-form data. GNN learns from nodes, edge, and graph structure. Basic GNN uses the message passing method where graph nodes iteratively update their representations by exchanging information with their neighbors [35]. However, early GNNs has a drawback in iterative message passing because it has proper optimization technique in aggregating node information. This led to computational inefficiency and difficulty in scaling to large graphs [34]. Graph convolution network (GCN) introduces the convolution calculation in the node information aggregation. GCN adopts a simple calculation of the adjacency matrix and node features to aggregate information from neighbors to represent the node [33].

## Proposed approach

In this study, we propose a model that uses four different adjacency matrices based on four types of social activities: trade records, guild membership, party records, and raid (large-scale party) records. At each time step, these matrices are created and used to construct four graphs. Each of these graphs is embedded through GCN layers, concatenated, and integrated into a unified representation. The integrated graphs generated at each time step are used as inputs to a Bi-LSTM network to identify the temporal patterns of social graph changes. These inputs subsequently pass through dense layers, where a neural ODE is applied, and output the churn or retention status of each user. An additional model without an ODE segment is developed and evaluated for comparison. 1. illustrates the proposed model.

At each timestamp $T_n$, dynamic graphs are processed through GCN layers to generate node embeddings that encode both structural information and interaction dynamics. These embeddings are concatenated to form a unified representation for the timestamp, which serves as input to the Bi-LSTM layer. The Bi-LSTM captures temporal patterns by processing the sequence bidirectionally, enabling the model to account for both past and future context in user behavior. The output of the Bi-LSTM layer is then passed into the neural ordinary differential equation (ODE) segment. This segment refines the temporal embeddings by modeling them as a continuous dynamic system, effectively smoothing out abrupt changes and providing a more stable representation of user behavior over time. By solving differential equations numerically, the Neural ODE ensures that the transitions between timestamps are consistent with the underlying temporal dynamics, improving both prediction accuracy and stability. Finally, the refined embeddings are passed through dense layers to generate the final

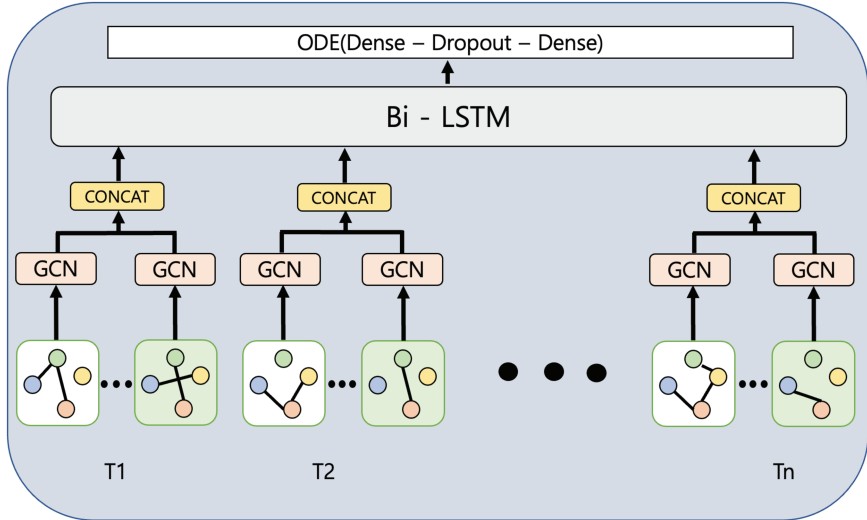

**Fig 1. Model structure of TempODEGraphNet.**

churn prediction, integrating information from both the dynamic graph structures and their temporal evolution.

## Social graph construction

We generate graphs based on various user interactions within the game, such as trade records, guild membership, party records, and raid records. A trade involves the exchange of goods, such as gold or items, between users. A guild refers to a group of individuals who gather for camaraderie or to share common goals or interests. A party is when up to 6 users form a single group to complete quests or engage in hunting activities. A raid extends this concept, where up to 4 parties systematically form a single group to handle high-difficulty dungeons.

In the trade graph, an edge is formed between two users when a transaction occurs between them. In the guild, party, and raid graphs, edges are formed between users who belonged to the same guild, party, or raid, respectively, during the data collection period. We add the following conditions for the three networks that can consider the weight of an edge to construct a strong edge.

- For creating edges in the trade graph, we exclude one-time transactions by considering only those instances where the same user IDs have conducted transactions at least twice on the same day and both user IDs belong to the sample of 10,000 users collected for the experiment.
- To exclude random party matching in the party graph, we consider only those instances where users have been in the same party at least twice on the same day or where the party's duration from creation to disbandment is at least 60 min.
- To exclude random party matching in the raid graph, we consider only those instances where users have been in the same party at least twice on the same day and have entered a raid dungeon at the same time.

## TempGraphNet

Our model applies a GCN to strengthen the influence among users who frequently interact and weaken it among those who do not. Similar to how convolutional neural networks (CNNs), widely used for image recognition, identify patterns by analyzing small, local regions of an image, GCNs analyze the local neighborhoods within a social network graph. In CNNs, a "kernel" (or filter) slides across the image, looking at small groups of pixels at a time and combining their values to detect features like edges or textures. This process creates "feature maps" that highlight these detected features. GCNs perform a similar process, but instead of pixels on a grid, they operate on nodes in a graph. Each node represents a user, and the connections (edges) represent their relationships. A GCN layer gathers information from a user's direct connections (friends, followers, etc.) and combines it to create a new representation for that user, capturing the influence of their social connections. Like CNNs emphasize local visual patterns, GCNs emphasize local social patterns within the network.

The mathematical operation of GCNs is as follows: Given a graph, $G = (V, E)$, where $V$ is the set of nodes and $E$ is the set of edges, each node, $v \in V$, has a feature vector, $x_v$. A GCN layer uses the input feature vector, $X \in \mathbb{R}^{N \times F}$, ($N$ is the number of nodes and $F$ is the number of features) and the adjacency matrix, $A$, of the graph to generate the output feature vector, $H$. This process is generally expressed as follows:

$$H = \sigma\left(\tilde{D}^{-1/2} \cdot \tilde{A} \tilde{D}^{-1/2} \cdot X \cdot W\right) \tag{1}$$

Here, $\tilde{A} = A + I$ is the adjacency matrix with added self-loops and $\tilde{D}$ is the diagonal node degree matrix of $\tilde{A}$. $W$ is a trainable weight matrix, and $\sigma$ is an activation function. We utilize trade, guild, party, and raid graphs, each of which passes through a GCN layer. Each graph represents a specific type of interaction among users, and the convolution operations on each graph generate embeddings that reflect the interaction information among users. Specifically, for each graph, $G_k$ ($k$ denotes the type of graph), we perform the following steps:

1. Prepare adjacency matrix $A_k$ and feature matrix $X_k$ for $G_k$.
2. Generate new feature matrix $H_k$ from $A_k$ and $X_k$ through the GCN layer.
3. Concatenate the generated feature matrices, $H_k$, across channels to create composite embedding $H$ for each user.

$$H = \text{concat}(H_1, H_2, H_3, H_4) \tag{2}$$

The reasons for concatenating the embeddings from each graph are as follows: First, each graph is generated based on different types of social activities, thereby enabling them to be processed independently. This ensures that each graph captures the unique patterns of social interactions through the independent application of GCNs.

Second, this method enables multimodal integration. Each graph represents different modalities, and a unified embedding is created by concatenating the graphs. This enables the model to comprehensively utilize diverse interaction information, thus facilitating a nuanced understanding of complex relationships and interaction patterns among customers.

Last, individual characteristics can be clearly represented by concatenating the embeddings from each graph. Each customer engages in various social activities, resulting in distinct interaction patterns. The model can accurately capture and model diverse relationships among customers by integrating the embeddings from multiple graphs, thereby enhancing the predictive performance of customer churn.

Therefore, the concatenation of the embeddings from each graph plays a crucial role in enhancing the accuracy and explanatory capability of the model through the independent processing and integration of multimodal interaction information.

Our model incorporates a Bi-LSTM layer based on two factors. First, we incorporate the Bi-LSTM layer to effectively learn the change patterns in user social graphs over time. Bidirectional Long Short-Term Memory networks (Bi-LSTMs) are a specialized type of recurrent neural network (RNN) designed for processing sequential data, such as time series. Unlike traditional RNNs, which process information in a single direction, Bi-LSTMs process input sequences in both forward and backward directions. This bidirectional processing enables the network to capture contextual information from both preceding and subsequent elements in the sequence. In the context of social network analysis, this means the Bi-LSTM can consider both past interactions and potential future trends to better understand the evolution of user behavior. This capability is crucial for accurately predicting user churn, as user behavior is influenced by both past actions and anticipated future interactions. Research on user churn prediction has highlighted the crucial role of capturing the evolution of user behavior over time [28,29]. Despite the necessity of designing domain-specific churn prediction models, the temporal evolution of user behavior remains a key factor in predicting user churn. The proposed model uses the Bi-LSTM layer to effectively capture the behavior change patterns of retained and churned users.

The Bi-LSTM layer processes sequences bidirectionally by simultaneously utilizing forward and backward LSTM. A Bi-LSTM consists of two LSTM networks: a forward LSTM and a backward LSTM. The forward LSTM processes the input sequence from the beginning to the end, while the backward LSTM processes the same sequence in reverse order. At each time step, the outputs from both the forward and backward LSTMs are combined, typically through concatenation. This combined output represents a more comprehensive understanding of the current element in the sequence, as it incorporates information from both its past and its future context. In our model, the input sequence to the Bi-LSTM is the series of user social graph embeddings generated by the GCN layers. By processing this sequence bidirectionally, the Bi-LSTM captures the dynamic changes in user relationships and behaviors over time. Specifically, for a given input sequence, $\mathbf{x} = (x_1, x_2, ..., x_T)$, the forward LSTM computes $\overrightarrow{\mathbf{h}}_t$ and the backward LSTM computes $\overleftarrow{\mathbf{h}}_t$, which are concatenated to form the final output, $\mathbf{h}_t = [\overrightarrow{\mathbf{h}}_t; \overleftarrow{\mathbf{h}}_t]$. This enables the model to learn the complex features of user behavior patterns at each timestamp.

Second, in contrast to previous studies that aggregated all interactions that occurred during the data collection period into a single graph, our approach considers the dynamic nature of trade targets, party members, guild affiliations, and other features within online games. These attributes can vary with timestamps. Therefore, we generate graphs at each time step to capture these dynamics, thus creating inputs in the form of dynamic graphs. This strengthens the influence among customers who interact with each other and incorporates temporal changes.

At each time step, the generated graphs pass through GCN layers, and the concatenated embeddings are used as inputs to the Bi-LSTM layer. This approach enables us to effectively model and analyze temporal changes while leveraging the aggregated influence among interacting users.

This approach consists of the following steps:

1. At each timestamp, $T_t$, the graph data are passed through GCN layers to generate embeddings for each node.

2. The generated embeddings are concatenated across channels to obtain a comprehensive embedding for that timestamp.

3. This sequence of comprehensive embeddings over time is used as the input to the Bi-LSTM layer to learn the changes in user behavior patterns.

Through this method, the proposed model effectively integrates the graph embeddings that reflect relationships among users at each timestamp while capturing temporal changes to enhance the accuracy of user churn prediction. Ultimately, the output of the Bi-LSTM layers is fed into the ODE segment to obtain refined predictions.

## TempODEGraphNet

Our model employs a neural ODE to enhance the consistency of model performance [30]. A neural ODE combines neural networks with differential equations; this makes it particularly useful for modeling time-series data or continuous dynamic systems. Conventional neural networks are used to learn complex relationships between inputs and outputs. They are composed of multiple layers of neurons that adjust weights and apply activation functions to generate outputs as data pass through a network. In contrast, a neural ODE models dynamic systems where the network output varies discretely, unlike conventional neural networks that generate fixed outputs for given inputs. This dynamic system is represented by the following differential equation:

$$\frac{d}{dt}\mathbf{x}(t) = f(\mathbf{x}(t), t) \tag{3}$$

Here, $\mathbf{x}(t)$ represents a function of time $t$ and $f(\mathbf{x}(t), t)$ defines the rate of change in $\mathbf{x}$. The function $f(\mathbf{x}(t), t)$ is modeled by a neural network for given input $\mathbf{x}$ and time $t$ to determine $f(\mathbf{x}(t), t)$ as a function of time in the network output. The model trained using the neural ODE starts from given initial conditions, and it predicts the state of the system over time through differential equations. Thus, it can be used to model dynamic systems.

The core idea of the neural ODE is to find the solution to the differential equation rather than directly update network parameters. For this purpose, differential equations are solved using numerical methods such as the Runge–Kutta method. The Dopri-5 method used in this study is a high-order Runge–Kutta method that provides high accuracy while considering computational efficiency. This method computes one 5th order approximation and another 4th order approximation, estimates the error by comparing them, and accordingly adjusts the step size. It is represented by the following formula:

$$y_{n+1} = y_n + \sum_{i=1}^{s} b_i k_i \tag{4}$$

- $y_{n+1}$: the approximated solution at the next time step.
- $y_n$: the current solution.
- $\sum_{i=1}^{s} b_i k_i$: the weighted sum of the gradients used to compute the next value of $y$, where $s$ is the number of stages in the Runge-Kutta method.
- $k_i$: the gradients calculated at each stage of the step, used to estimate the slope (derivative) of the solution curve at different points within the current step.
- $h$: the step size.
- $t_n$: the current time.
- $c_i$: the coefficients that determine the points within the interval $[t_n, t_n + h]$ at which the function evaluations are to be made.

- $a_{ij}$: the coefficients used to weigh the gradients from previous stages for calculating each subsequent gradient $k_i$.

Here, $\mathbf{k}_i$ represents the gradient at each step, and it is calculated as follows:

$$
\begin{aligned}
k_1 &= hf(t_n, y_n) \\
k_2 &= hf(t_n + c_2 h, y_n + a_{21} k_1) \\
k_3 &= hf(t_n + c_3 h, y_n + a_{31} k_1 + a_{32} k_2) \\
&\vdots \\
k_s &= hf(t_n + c_s h, y_n + \sum_{j=1}^{s-1} a_{sj} k_j)
\end{aligned}
\tag{5}
$$

The Dopri-5 method efficiently solves differential equations within specified error tolerances by dynamically adjusting step sizes using the 5th and 4th order approximations at each step. This method enables neural ODEs to simultaneously maintain high accuracy and stability. Conventional neural networks directly learn the mapping between inputs and outputs. The gradient descent method is used to update the weight that connects inputs and outputs in the direction that reduces the loss. The gradients required to calculate the weight update are obtained using backpropagation. In contrast, neural ODEs find solutions to differential equations starting from given initial conditions, thereby modeling the dynamic changes in systems. This enables neural ODEs to approximate primitive functions more effectively compared with conventional methods. Neural ODEs achieve sophisticated predictions while maintaining model coherence. This is particularly beneficial for addressing dynamic and complex patterns such as customer churn prediction.

## Experiments

Fig 2 shows the table that visually represents the process of analyzing user activity data to determine churn (user dropout) and retention (continued activity). The table is structured along two axes. X-axis (horizontal) represents time in weeks. The training preiod (weeks 1–6) includes the data collection phase for training the model, where user activity is recorded. The no data period (weeks 7–9) is employed as a gap period where no user activity data is collected. The observation period (week 10 onward) is set the phase where user behavior is analyzed to classify them as either churned or retained. Y-axis (vertical) lists some individual users (User A to User F). Black dots and lines indicate user activity at specific time points. The users with brown color are classified into the churned users who stop activity after week 10. The users with blue color are retained users who continue activity beyond week 10. For example, user A was active until week 6 but showed no activity in week 10 and beyond, classifying them as a churned user. In contrast, user D remained active beyond week 10, making them a retained user.

This study utilizes data collected from 10,000 users of the MMORPG game "Blade & Soul" (NCSOFT). Less 30% users of total users were randomly selected after malicious users and inactive users were excluded. This data was made available during the GDMC 2017 competition, which featured two tracks. Both Track 1 (customer churn prediction), which is the focus of this study, and Track 2 used the same dataset of 10,000 users. This dataset is divided into three subsets collected over different periods: Train, Test1, and Test2. Across the entire dataset of 10,000 users, 30% are churn users (defined as not exhibiting activity during a designated

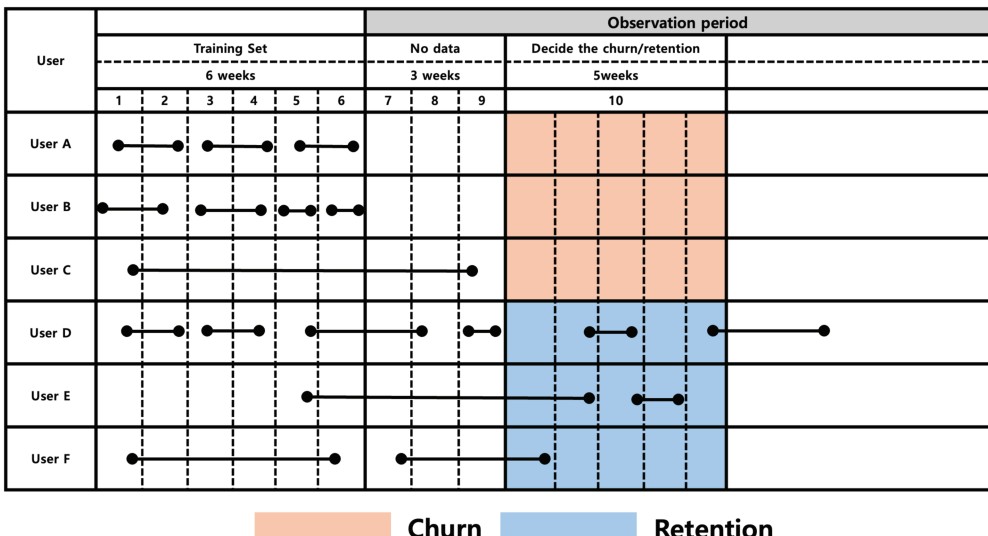

**Fig 2. Criteria for defining customer churn.**

churn window despite having prior activity). The basic details of the dataset are provided in Table 2. The dataset can be accessed at https://doi.org/10.6084/m9.figshare.28233857.v1.

'Blade & Soul' is one of Korea's representative MMORPGs. It is renowned for its unique fusion of martial arts and fantasy elements, coupled with an action-oriented combat system. Based on a high degree of freedom, it provides various gameplay modes where players can enjoy player-versus-player battles and cooperative dungeon clears. Users can form guilds and parties, engage in trading, and participate in a wide range of activities. The provided data encompass 82 major types of logs that can be collected within the game. These logs capture information related to server connections, characters, items, skills, quests, and guilds. Each log type is structured into 77 fields categorized into common, actor, object, and target fields. Common fields include universal information recorded for all log types, such as the log type, creation time, and in-game location of an action. Object fields contain details about the subject of the action, including item IDs, item grades, item quantities, and skill names. Target fields include information about the target of the action, such as the character IDs that are the targets in a trade and the character IDs invited to a party.

To investigate the differences in behavior between churned and retained users, we conducted exploratory data analysis (EDA) on the entire dataset of 10,000 users. This analysis aimed to establish criteria for graph generation based on these distinct interaction patterns within the game. This analysis is performed on a learning dataset collected over 6 weeks from 4,000 participants. Each feature is defined using the interquartile range (IQR), which is calculated as Q3 - Q1. We remove outliers by selecting the values that are greater than or equal to

**Table 2. Dataset information.**

| Dataset | Time | Weeks | #Users |
|---|---|---|---|
| Train | 04-01-2016 ~05-11-2016 | 6 | 4000, 30% Churn |
| Test1 | 06-27-2016 ~09-21-2016 | 8 | 3000, 30% Churn |
| Test2 | 12-14-2016 ~02-08-2017 | 8 | 3000, 30% Churn |

Q1 - 1.5 × IQR and less than or equal to Q3 + 1.5 × IQR. We compared the playtime, experience points (Exp) earned, mastery Exp earned, in-game currency obtained through hunting, quest rewards, and player trades, currency spent, and party participation frequency between churned users and retained users. The results are shown in Fig 3.

For playtime, retained users had longer playtimes compared to churned users. On the other hand, churned users earned more Exp than retained users. This is attributed to the system characteristics of 'Blade & Soul.' At the time of data collection, the maximum level in 'Blade & Soul' was 55. Players earn Exp through quest and hunting rewards until they reach level 55. After reaching level 55, players earn mastery Exp instead of regular Exp. Comparing Exp and mastery Exp, it was found that a higher proportion of retained users had reached the maximum level. Moreover, they continued to actively participate in hunting, quests, and other content even after reaching the maximum level. When comparing the changes in currency obtained through player trades, hunting, and quest rewards, both currency earnings and expenditures were higher for retained users. Additionally, retained users had more frequent party participation, which is essential for completing high-difficulty dungeons. The EDA results indicated that retained users played the game for longer periods, continued to engage in hunting, quests, party hunting, and raids even after reaching the maximum level, and actively utilized the party system. The currency variations from hunting rewards, quest rewards, NPC trades, and player trades were also higher for retained users compared

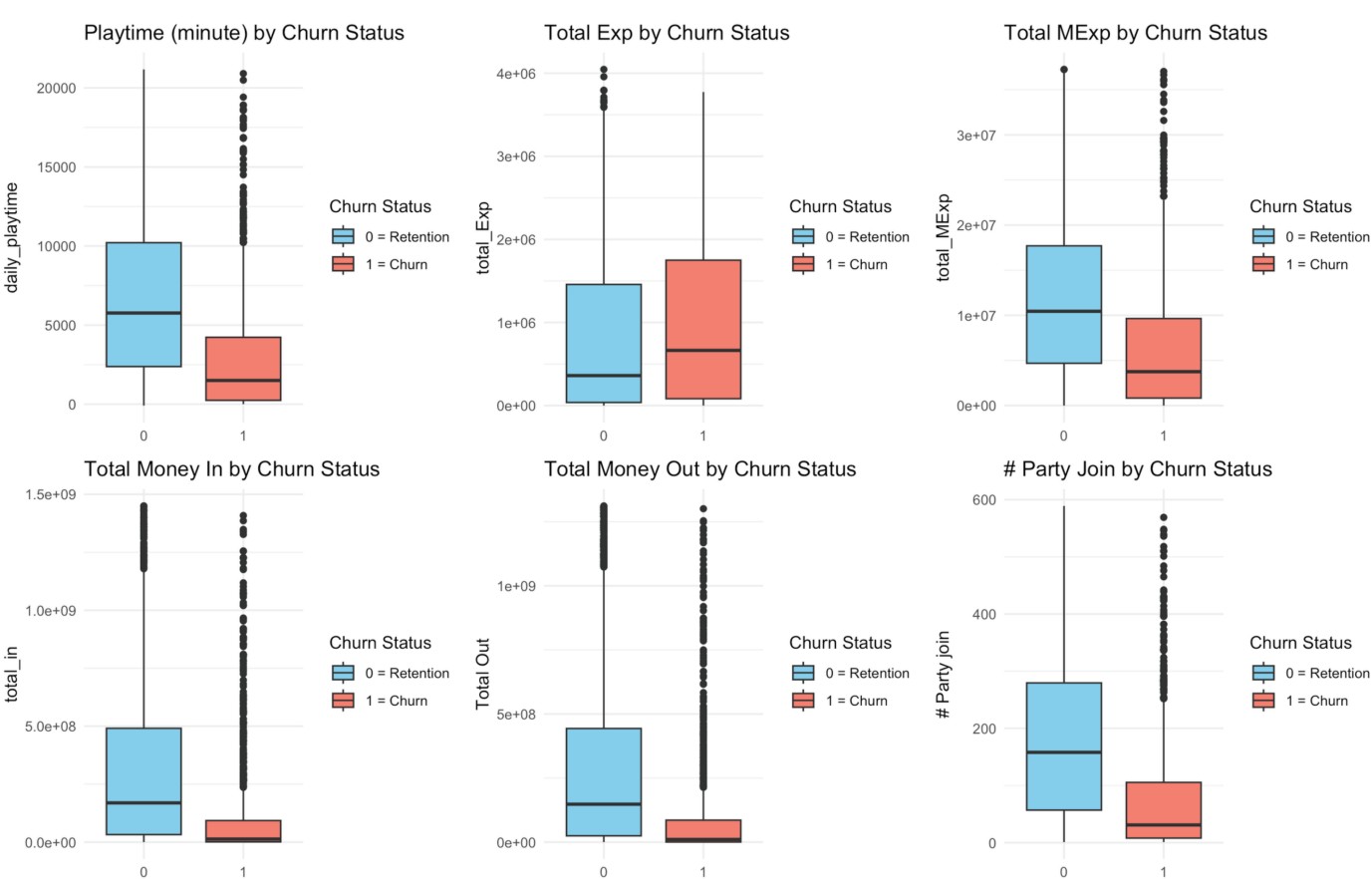

**Fig 3. Comparison of features between churned and retained users.**

to churned users. Considering these results, we selected user interaction records, including player trades, party records for XP acquisition, large-scale party raids, and guild memberships, as the basis for generating graphs.

Forty six features are generated using this dataset; these include the playtime, experience gained, and number of trades. Principal component analysis (PCA) is performed based on these generated features, and 6 principal components that collectively explain over 85% of the variance are selected as shown in Table 3. PCA reduces the number of features while preserving essential information, improves data quality by removing the features that are unrelated to significant variations or noise, and enhances computational efficiency by decreasing the dimensionality. In the PCA results, the first principal component has a coefficient of 0.294 for the playtime. This directly reflects user engagement in the game. The total experience gained and total mastery have coefficients of 0.204 and 0.19, respectively. Both of these features are significant indicators of user achievement within the game. Participation in parties has a coefficient of 0.261, and it indicates the level of social activity of users. The features related to resurrection and NPC kills have coefficients of 0.224 and 0.202 respectively, and they reflect user combat activity. These features collectively capture various aspects of user gameplay activity and play crucial roles in predicting user behavior patterns and churn likelihood. Specifically, the playtime and experience metrics are key indicators of user engagement and achievement. Participation in parties and resurrection frequency are important metrics for social engagement and survival capability. The key fatures are listed in Table 4. The performance of the model is optimized by utilizing these features.

**Table 3. PCA cumulative proportion and key features for each component.**

| Component | Cumulative Proportion | Key Features |
|---|---|---|
| PC1 | 0.3425 | playtime, total_Exp, party_join |
| PC2 | 0.5015 | total_currencyIn, total_currencyOut, kill_npc |
| PC3 | 0.6365 | death, kill_npc, npc_battle |
| PC4 | 0.7285 | revive, death, enter_pvpServer |
| PC5 | 0.8000 | near_death, total_Exp, revive |
| PC6 | 0.8625 | near_death, trade_give, kill_player |

**Table 4. Important features according to feature groups.**

| Game engagement | |
|---|---|
| playtime | Each user's play time |
| sessions | Number of sessions user plays |
| total_currency_I/O | Daily currency fluctuation for user |
| **Game achievement** | |
| total_Exp | Total acquired Exp through hunting, quest rewards, etc. |
| total_MExp | Total acquired mastery Exp |
| **Social activity** | |
| party_join | Number of times a user participates in the party |
| party_invite | Number of invitations to a user's party |
| trade_give \| receive | Number of trade between users |
| **Combat activity** | |
| kill_npc | Number of neural user character kills |
| kill_player | Number of other user character kills |
| revive | Number of user character resurrections (= Number of deaths) |

## Experimental results

The performance of the model is measured as the average of the F1 scores for both test sets, and the F1 scores are defined as follows:

$$F1 = 2 \times \frac{\text{precision} \times \text{recall}}{\text{precision} + \text{recall}} \tag{6}$$

The performance of the proposed model is compared with that of the top-performing participant in Track 1 of the GDMC 2017 competition using the same dataset [31]. The Yokozuma Data team, which achieved the highest ranking in GDMC 2017 Track 1, addressed the data imbalance issue through cost sensitive learning and adjusted label strategies. They utilized nonlinear modeling to determine temporal patterns. For structural modeling based on decision trees, they employed preprocessing techniques, such as linear discriminant analysis and PCA, to extract an optimal feature set. They optimized their model using the XGBoost algorithm and hyperopt and achieved an average F1 score of 0.62 across Test1 and Test2. The performance comparison evaluates the impact of considering social interaction on various models, including TempODEGraphNet , tempGraphNet (TGN) trained using standard methods, Yokozuma Data's model, decision trees, random forests, and static GCN models. The results are presented in Table 5.

To ensure consistency in model training, tempODEGraphNet and TGN are input in chronological order rather than random order. The Adam optimizer with a learning rate of 0.001 is used. Training is performed for 1000 epochs with early stopping.

The proposed model demonstrates the best performance in terms of the average F1 score, regardless of whether the neural ODE is applied. The performance gap between TempODE-GraphNet and static graph models highlights the limitations of static approaches in dynamic settings. Static Graph Convolutional Networks has limitations in that they aggregate user interactions over the entire data collection period without accounting for temporal changes. This approach fixes node states and edge connections, ignoring the dynamic nature of user behavior. Interactions (edges) between users that exist at one point in time may disappear or reemerge at another, reflecting fluctuations in graph data. These temporal variations are important for detecting early signals of user churn. By reducing these interactions into a single static graph, important temporal patterns are smoothed out or lost, which can result in delayed churn detection or the misclassification of active users as retained. TempODEGraph-Net addresses this limitation by constructing dynamic graphs at each timestamp, allowing

**Table 5. Performance comparison.**

| Model | Dataset | Accuracy | Precision | Recall | F1 score | Average F1 score |
|---|---|---|---|---|---|---|
| Ours (TempODEGraphNet) | Test1 | 0.77 | 0.59 | 0.73 | 0.65 | 0.65 |
| | Test2 | 0.75 | 0.57 | 0.73 | 0.64 | |
| TGN | Test1 | 0.76 | 0.58 | 0.74 | 0.65 | 0.64 |
| | Test2 | 0.75 | 0.56 | 0.72 | 0.63 | |
| Yokozuma's [31] | Test1 | 0.74 | 0.55 | 0.69 | 0.61 | 0.62 |
| | Test2 | 0.73 | 0.54 | 0.76 | 0.63 | |
| Decision Tree | Test1 | 0.58 | 0.34 | 0.41 | 0.37 | 0.43 |
| | Test2 | 0.61 | 0.40 | 0.61 | 0.48 | |
| Random Forest | Test1 | 0.71 | 0.51 | 0.62 | 0.56 | 0.57 |
| | Test2 | 0.70 | 0.50 | 0.68 | 0.58 | |
| Static GCN [25] | Test1 | 0.73 | 0.54 | 0.67 | 0.60 | 0.60 |
| | Test2 | 0.73 | 0.54 | 0.70 | 0.61 | |

the model to track changes in node states and edge connections over time. This ensures that short-term fluctuations and evolving user relationships are effectively captured, providing a more comprehensive representation of user behavior in dynamic environments. In addition, the performance of the static GCN is better than that of random forests for the same feature matrix. This suggests that considering interactions between customers positively improves the model performance for predicting customer churn, particularly in the domain of MMORPGs. To further analyze the model performance, confusion matrices for ours and Yokozuma's models are presented in Table 6.

The confusion matrices highlight key differences in the models' predictive capabilities. TempODEGraphNet demonstrates consistently lower false positive (FP) and false negative (FN) rates across both datasets, indicating superior model stability and reliability. Specifically, TempODEGraphNet achieves lower FP rates compared to Yokozuma's model. This suggests that TempODEGraphNet is more conservative in predicting churn, resulting in fewer incorrect churn predictions for users who were actually retained. Moreover, TempODEGraphNet records higher true positive values in Test1 compared to Yokozuma's model, demonstrating improved sensitivity to churn cases. However, Yokozuma's model achieves slightly higher TP values in Test2 compared to TempODEGraphNet, indicating that while Yokozuma's model may capture more churn users, it does so at the cost of increased false alarms (FP). These findings suggest that TempODEGraphNet offers a more balanced approach, effectively capturing churn cases while minimizing false positives, which can be critical in applications where overestimating churn could lead to unnecessary interventions or resource misallocations.

To compare the model performance based on the application of the neural ODE, we measure the performance by randomly removing 10 days from the 6 week training dataset and varying the random seed. The results are shown in Table 7.

**Table 6. Confusion Matrices for Ours and Yokozuma's Model on Test1 and Test2.**

| Test1 Confusion Matrix | | | | |
|---|---|---|---|---|
| **Model** | **TN** | **FP** | **FN** | **TP** |
| Ours | 1645 | 455 | 243 | 657 |
| Yokozuma's | 1591 | 509 | 279 | 621 |
| **Test2 Confusion Matrix** | | | | |
| TGN-N | 1601 | 499 | 243 | 657 |
| Yokozuma's | 1537 | 563 | 216 | 684 |

**Table 7. Performance comparison based on the application of neural ODE in terms of F1 score.**

| Trial | Test1 | Test2 | Average | Test1 | Test2 | Average |
|---|---|---|---|---|---|---|
| - | **Neural ODE** | | | **Conventional** | | |
| 1 | 0.60 | 0.60 | 0.60 | 0.59 | 0.61 | 0.60 |
| 2 | 0.60 | 0.59 | 0.60 | 0.54 | 0.58 | 0.56 |
| 3 | 0.58 | 0.61 | 0.60 | 0.54 | 0.60 | 0.57 |
| 4 | 0.59 | 0.62 | 0.61 | 0.64 | 0.65 | 0.65 |
| 5 | 0.58 | 0.59 | 0.59 | 0.60 | 0.60 | 0.60 |
| 6 | 0.61 | 0.60 | 0.61 | 0.57 | 0.60 | 0.59 |
| 7 | 0.60 | 0.58 | 0.59 | 0.60 | 0.60 | 0.60 |
| 8 | 0.61 | 0.58 | 0.60 | 0.55 | 0.54 | 0.55 |
| 9 | 0.59 | 0.60 | 0.60 | 0.64 | 0.66 | 0.65 |
| 10 | 0.60 | 0.57 | 0.59 | 0.60 | 0.58 | 0.59 |
| Mean | 0.60 | 0.59 | 0.60 | 0.58 | 0.60 | 0.59 |
| Std. | 0.011 | 0.015 | - | 0.044 | 0.033 | - |

**Table 8. Performance comparison results.**

| Test1 | | |
|---|---|---|
| Indicator | Neural ODE | Conventional |
| Mean | 0.599 | 0.578 |
| Variance | 0.0001252 | 0.0027536 |
| Observations | 20 | 20 |
| df | 19 | 19 |
| P($F \leq f$) one-tail | 3.93E-09 | - |
| Test2 | | |
| Indicator | Neural ODE | Conventional |
| Mean | 0.597 | 0.608 |
| Variance | 0.0001986 | 0.0007852 |
| Observations | 20 | 20 |
| df | 19 | 19 |
| P($F \leq f$) one-tail | 0.0021714 | - |

The model with the neural ODE achieves an average F1 score of 0.60, which is comparable to the F1 score of the conventional model (0.59). However, the standard deviation (Std.) of the model with the neural ODE is smaller (0.011 for Test1, 0.015 for Test2, and 0.007 overall) than that of the conventional model (0.044 for Test1, 0.033 for Test2, and 0.035 overall). This indicates that the neural ODE enhances the consistency and stability of the model.

To statistically assess whether there is a difference in the consistency and stability of the models with and without the neural ODE, an F-test is conducted to verify if the variances between the two groups differ significantly. Performance metrics are recorded up to trial 20 for the models with the neural ODE and conventional models on the Test1 and Test2 datasets. Separate F-tests are performed for comparing the performances of the models for the Test1 and Test2 datasets. The results are presented in Table 8. The F-test results reveal that the p-values for Test1 ($3.93 \times 10^{-9}$) and Test2 (0.0021714) are both below the 0.05 significance threshold. This indicates that the variances between the two groups are significantly different, implying that the model incorporating the neural ODE demonstrates significantly lower variability compared to the conventional model. The reduced variance reflects that the neural ODE enhances the stability and consistency of the model across different trials. While this analysis focuses on the stability of the model's performance, it supports the argument that the neural ODE contributes to more reliable and robust predictions by minimizing fluctuations in performance. This is particularly critical in dynamic environments where consistent outputs are essential for reliable churn prediction. The significant difference in variance suggests that the neural ODE's integration leads to a model that is less sensitive to variations in the data, reinforcing its advantage over conventional approaches.

## Conclusion

This study proposes a graph-based neural network model, referred to as TempODEGraphNet, for predicting customer churn in online games. The model reflects user interactions by calculating centrality metrics and applying Graph Convolutional Networks (GCNs), which capture the structural information of the network and graph.

Additionally, the model accounts for the domain-specific characteristic that the structure of the graph can change over time, reflecting the temporal evolution of user interactions.

The novelty of TempODEGraphNet lies in its application of dynamic graph structures to predict customer churn in online games and its pioneering use of Neural Ordinary Differential Equations (Neural ODEs) to enhance model performance consistency.

Another significant contribution of TempODEGraphNet is the configuration of multiple GCNs to learn the characteristics of various social networks within the game, recognizing patterns of change in these characteristics over time. Experimental results demonstrate that the proposed model outperforms existing models in terms of F1 score. This performance improvement can be attributed to the effective capture of temporal changes in user behavior using dynamic graphs, as well as the application of Bidirectional Long Short-Term Memory (Bi-LSTM) and Neural ODEs to enhance model learning performance.

In online game churn prediction, minimizing false positives (FP) and false negatives (FN) is crucial, as FP leads to unnecessary retention incentives for players unlikely to leave, while FN results in missed opportunities to prevent actual churn, directly impacting revenue [11]. The confusion matrices obtained from our experiments reveal that proposed model consistently achieves lower FP and FN rates compared to Yokozuma's model. This reduction in FP optimizes operational costs by reducing unnecessary resource allocation for retention efforts. More importantly, the lower FN rate enhances the identification of actual churners, mitigating revenue loss and improving player retention. These results highlight the practical value of the proposed model, not only in improving predictive accuracy but also in minimizing the financial impact associated with misclassification errors in churn prediction systems.

However, our study has certain limitations. The data used in the experiments are specific to one online game, which may affect the model's performance when applied to different games or domains. Nevertheless, we anticipate that the model can generalize well to other MMORPGs or similar environments with appropriate feature engineering and exploratory data analysis (EDA). By generating feature matrices that reflect the most distinguishing characteristics between churned and retained users and constructing adjacency matrices based on social activities such as in-game friendships and chat logs, it is possible to prepare inputs required for GCNs. This process can help adapt the model to various MMORPGs or similar domains. Futhermore, the proposed methodology can be applied to MMORPG games that allow the formation of social networks within the game. Furthermore, it is applicable not only to games but also to domains where social networks dynamically evolve over time. Following are possible application domains. Social network platforms like Twitter, Facebook, and Instagram show constant changes in user connections through actions such as follows, unfollows, conversations, comments, and likes. Online learning platform such as MOOC platforms (Massive Open Online Courses) shows dynamic networks as students join team projects or discussion groups.

Future research will focus on enhancing model interpretability by exploring methods to better understand the internal mechanisms of the model. Additionally, we plan to investigate the optimization of graph aggregation methods, considering that graphs formed under different network criteria may play varying roles in predicting customer churn. We aim to develop methods for dynamically determining the optimal aggregation ratios (mixture ratios) during model training. Finally, we intend to explore the application of reparameterization techniques, similar to those used in YOLO-v7, to further expand the application of ODE segments in the proposed model.

## Author contributions

**Supervision:** Jiyoung Woo.

**Writing – original draft:** Minseop Lee.

**Writing – review & editing:** Jiyoung Woo.

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
