## [Decision Letter · Decision Letter 0]

PONE-D-24-30725TempODEGraphNet: Predicting User Churn Using Dynamic Social Graphs and Neural ODEsPLOS ONE

Dear Dr. Woo,

Thank you for submitting your manuscript to PLOS ONE. After careful consideration, we feel that it has merit but does not fully meet PLOS ONE’s publication criteria as it currently stands. Therefore, we invite you to submit a revised version of the manuscript that addresses the points raised during the review process.

We look forward to receiving your revised manuscript.

Kind regards,

Sajid Anwar, Ph.D

Academic Editor

PLOS ONE

Journal Requirements:

3. Thank you for stating the following financial disclosure: Soonchunhyang University;  the Regional Innovation Strategy (RIS) through the National Research Foundation of Korea (NRF) by the Ministry of Education (MOE) under Grant 2021RIS-004 

Reviewers' comments:

Reviewer's Responses to Questions

**Comments to the Author**

1. Is the manuscript technically sound, and do the data support the conclusions?

Reviewer #1: Yes

Reviewer #2: Yes

2. Has the statistical analysis been performed appropriately and rigorously? 

Reviewer #1: N/A

Reviewer #2: Yes

3. Have the authors made all data underlying the findings in their manuscript fully available?

Reviewer #1: Yes

Reviewer #2: Yes

4. Is the manuscript presented in an intelligible fashion and written in standard English?

Reviewer #1: Yes

Reviewer #2: Yes

5. Review Comments to the Author

Reviewer #1: The manuscript presents an approach to predicting user churn in online games using dynamic social graphs and neural ordinary differential equations (ODEs). However, the manuscript need following modification to be considered for publication.

Clarity and Readability

• The manuscript frequently uses terms such as "graph convolutional networks (GCNs)" and "bidirectional LSTM (Bi-LSTM)" without providing basic explanations for these terms. For instance, when introducing GCNs (p. 54), a brief primer on their operation could help non-experts understand their relevance to dynamic social graphs.

• Figure 1 illustrates the model structure but lacks detailed annotations that explain each component's role, especially how inputs from dynamic graphs are processed through the neural ODEs.

Methodological

• The data sourced from 10,000 users of 'Blade & Soul' are mentioned (p. 34), but there is no discussion on how these data points are distributed or their characteristics.

• The manuscript compares the TempODEGraphNet model with "conventional algorithms and static graph models" (p. 44), but does not specify what these models lack in dynamic settings.

Experimental

• The paper states that "the proposed model achieves a higher F1 score compared with conventional algorithms" (p. 46) without providing statistical tests to support the significance of these differences. Include a table of p-values or confidence intervals for F1 scores to statistically validate that improvements are not due to random variations in data.

• Only the F1 score is used to evaluate model performance. Report additional metrics like accuracy, precision, recall, and AUCROC to give a complete picture of model performance across different thresholds and conditions.

Broader Implications and Generalizability:

• The model is tested exclusively on data from 'Blade & Soul'. Discuss preliminary tests on how the model might perform with data from other MMORPGs or similar domains, highlighting any adjustments needed for different types of user interaction data.

Improvement: Discuss or conduct preliminary tests on how the model might perform with data from other MMORPGs or similar domains, highlighting any adjustments needed for different types of user interaction data.

• The consequences of false positives (predicting churn where there is none) and false negatives (failing to predict actual churn) are not discussed. Analyse the impact of these misclassifications on the game's revenue or player engagement strategies, providing insights into how the model's predictions could be used in operational settings.

Final Note: Consider revising and resubmitting.

Reviewer #2: The paper “TempODEGraphNet: Predicting User Churn Using Dynamic Social Graphs and Neural ODEs” addresses the critical issue of churn prediction in the gaming industry through an innovative approach using Graph Neural Networks (GNNs) and dynamic social graphs. The research is well-structured, with a compelling and relevant topic that aligns with advancements in predictive modelling. The authors effectively contextualize their work by reviewing and integrating recent literature, though further elaboration on how their model extends existing methods would strengthen the introduction.

The use of dynamic graphs and neural ODEs to capture temporal interactions is a significant contribution. Experimental results are clearly presented, with a notable improvement in F1 scores, validating the proposed model's efficiency. The discussion of limitations and future directions is commendable, particularly the focus on interpretability and optimization of graph aggregation methods. However, the study's reliance on data from a single game limits the generalizability of findings. Overall, the paper provides a valuable contribution to churn prediction and dynamic graph modelling, with well-justified methods and results.

6. PLOS authors have the option to publish the peer review history of their article (what does this mean?). If published, this will include your full peer review and any attached files.

Reviewer #1: No

Reviewer #2: **Yes: **Dr Leonidas Katelaris

---

## [Author Response · Author response to Decision Letter 1]

19 Jan 2025

We are sincerely grateful for your thorough consideration and scrutiny of our manuscript, “TempODEGraphNet: Predicting User Churn Using Dynamic Social Graphs and Neural ODEs”, control number PONE-D-24-30725. Through the accurate comments made by the reviewers, we better understand the critical issues in this paper. We have revised the manuscript according to the Reviewer’s suggestions. We hope that our revised manuscript will be considered and accepted for publication in the PLOS ONE. We acknowledge that the scientific and statistical quality of our manuscript was improved by the scrutinizing efforts of the reviewers and editors.

The changes within the revised manuscript were highlighted in blue. Point-by-point responses to the reviewers’ comments are provided below.

---

## [Decision Letter · Decision Letter 1]

TempODEGraphNet: Predicting User Churn Using Dynamic Social Graphs and Neural ODEs

PONE-D-24-30725R1

Dear Dr. Woo,

We’re pleased to inform you that your manuscript has been judged scientifically suitable for publication and will be formally accepted for publication once it meets all outstanding technical requirements.

Kind regards,

Sajid Anwar, Ph.D

Academic Editor

PLOS ONE

Additional Editor Comments (optional):

Reviewers' comments:

Reviewer's Responses to Questions

**Comments to the Author**

1. If the authors have adequately addressed your comments raised in a previous round of review and you feel that this manuscript is now acceptable for publication, you may indicate that here to bypass the “Comments to the Author” section, enter your conflict of interest statement in the “Confidential to Editor” section, and submit your "Accept" recommendation.

Reviewer #1: All comments have been addressed

Reviewer #2: All comments have been addressed

2. Is the manuscript technically sound, and do the data support the conclusions?

Reviewer #1: Yes

Reviewer #2: Yes

3. Has the statistical analysis been performed appropriately and rigorously? 

Reviewer #1: N/A

Reviewer #2: N/A

4. Have the authors made all data underlying the findings in their manuscript fully available?

Reviewer #1: No

Reviewer #2: Yes

5. Is the manuscript presented in an intelligible fashion and written in standard English?

Reviewer #1: Yes

Reviewer #2: Yes

6. Review Comments to the Author

Reviewer #1: All my comments have been addressed. I would like to accept the paper in the current form. Thank you

Reviewer #2: This paper proposes TempODEGraphNet, a model that aims at predicting user churn in gaming by using dynamic social graphs and Neural ODEs. The study is also well organised and has been aligned to recent literature, and it contributes to the field by considering temporal interactions which lead to better F1 scores. Although the introduction could fail to clearly explain how the model is different from the existing methods and using data from a single game may hamper the generalizability of the findings, the work is still a useful addition to the churn prediction and dynamic graph modelling literature. Moreover, the authors have responded to all the reviewer comments.

7. PLOS authors have the option to publish the peer review history of their article (what does this mean?). If published, this will include your full peer review and any attached files.

Reviewer #1: No

Reviewer #2: No

---

## [Editor Report · Acceptance letter]

PONE-D-24-30725R1

PLOS ONE

Dear Dr. Woo,

I'm pleased to inform you that your manuscript has been deemed suitable for publication in PLOS ONE. Congratulations! Your manuscript is now being handed over to our production team.

Kind regards,

on behalf of

Dr. Sajid Anwar

Academic Editor

PLOS ONE